# GramStatTexNet: Efficient, Interpretable, and Neuro-Inspired Spatiotemporal Texture

## Abstract

The development of sophisticated texture modeling and synthesis techniques, combined with deep connections to human vision modeling, has propelled advances in visual neuroscience, computer graphics, and beyond. Human peripheral vision is well modeled as local texture, scaled by distance from the center of gaze, with the most highly human validated models utilizing biologically-inspired filters and hand-curated statistics sets. Such models offer clear interpretability and a strong biological basis, but suffer from speed limitations and an inability to extend beyond the spatial domain. Conversely, deep learning methods like style-transfer and diffusion models generate high-quality results but at the cost of interpretability, biological plausibility, and fine-grain control, and are highly over-parameterized. We introduce GramStatTexNet, an analysis-by-synthesis model combining the multi-scale Gabor filter structure of classical texture models with the power and flexibility of Grammian-based approaches. Our model generates texture syntheses with similar quality to deep learning models while remaining interpretable, efficient, and biologically inspired. We create an organizational structure for our model statistics and leverage contrastive learning to identify statistics most important for categorizing texture, showing that this ordering correlates with synthesis quality, and identifying a further reduced set of statistics that retains high-quality synthesis. We demonstrate the tiled application of our model to full images, aggregating statistics over spatially-varying regions, an extension necessary for synthesizing foveated mongrels/metamers. In addition, we use our method to extend synthesis into the spatiotemporal domain with videos, paving the way for spatiotemporal peripheral vision models. Finally, we explore the incorporation of our statistics into modern diffusion models using gradient guidance. Our work bridges the gap between interpretability and high performance for texture models, providing an efficient framework for modeling human visual perception across space, time, and gaze location.

## 1 Introduction

The human visual system demonstrates state-of-the art performance and efficiency at processing and interpreting the dynamic visual world. In particular, human vision has the ability to recognize complex textures that are essential for identifying objects, surfaces, and materials. Humans achieve this both for static scenes and in the context of motion, enabling us to navigate and interact seamlessly with our environment. Understanding how to model these capabilities is crucial for advancing computer vision and machine learning applications in embodied and physical AI where we must interact with real-world environments. These goals are also closely related to interpretability, where biological-vision can as a natural source of inspiration for learning effective and more transparent visual representations.

Neuroscience-informed approaches to texture synthesis (and closely related peripheral vision) often model visual areas using non-parametric multi-scale pyramids, decomposing images into filter responses, and aggregating features into statistics sets (Portilla and Simoncelli, 2000). These statistics sets are predictive of neural responses to natural scenes (Ziemba et al., 2016), produce perceptually plausible textures, and serve as models of peripheral vision when applied over spatial pooling regions (Rosenholtz et al. (2012); Freeman and Simoncelli (2011)). Despite these advantages for images, these texture models typically struggle to capture spatiotemporal dynamics. Many of the

spatial statistics they rely on do not have clear analogues in the temporal domain, and humans are particularly sensitive to temporal artifacts that can arise during rendering. Deep learning approaches to generating spatial textures, however, suggest that there are alternative and more principled approaches to feature selection that could be used to help extend neuroscience-informed models into the spatiotemporal domain.

Deep-learning based approaches to texture and peripheral synthesis offer strong results in terms of quality and speed, but their reliance on over-parameterized latent spaces comes at the cost of interpretability and biologically feasibility. Style-transfer-based approaches synthesize textures by matching correlations between the hidden layers of both deep Gatys et al. (2015) and single-layer (Ustyuzhaninov et al., 2017) convolutional networks. This approach has also shown success for peripheral syntheses (Wallis et al., 2017; Deza et al., 2017). However, these models rely on pretrained layers from networks like VGG (Simonyan and Zisserman, 2014), and thus offer no representational compression compared to the highly efficient texture models from visual neuroscience.

In this work, we introduce GramStatTexNet, a novel approach that bridges the gap between interpretability, biological-plausibility, and high performance in texture and peripheral vision modeling. Our analysis-by-synthesis approach generates high-quality texture images comparable to deep network-based texture models, while remaining efficient and interpretable. We organize statistical representations into families and use contrastive learning to identify the most critical statistics for texture differentiation, allowing us to significantly reduce parameters without compromising synthesis quality. Additionally, we extend our model to produce full-frame peripheral vision syntheses (mongrels/metamers) through spatial pooling, and further apply it to videos, modeling human visual perception across space, time, and gaze location. Finally, we explore the ability of various diffusion models to generate textures that reflect these statistics through gradient guidance. Our framework unites the interpretability of classical models with the flexibility of deep learning, advancing efficient modeling of human visual perception in both spatial and temporal contexts.

## 2 PREVIOUS WORK

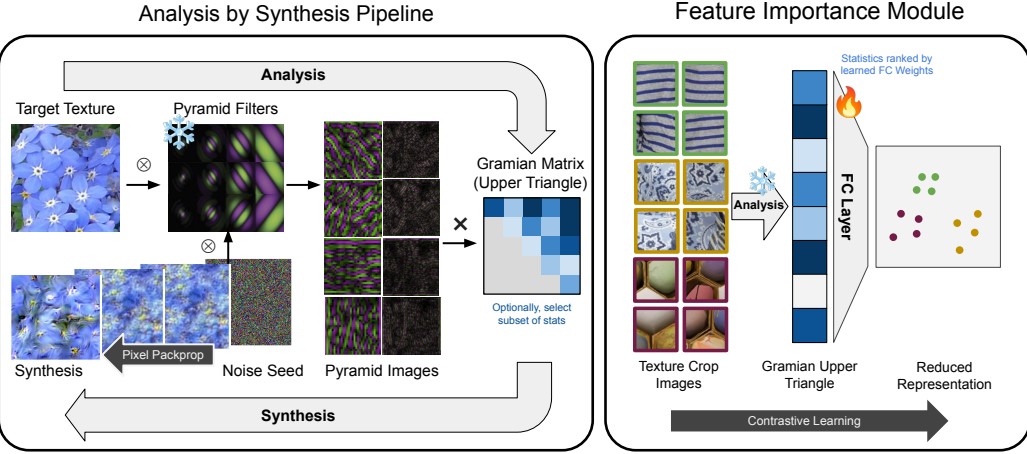

Figure 1: Left: Our analysis-by-synthesis texture model decomposes images into stacks (pyramids) of varying scales, orientations, and color channels, using with a multi-scale pyramid filter bank. Pair-wise correlations between all pyramid images are represented in the gram matrix; the upper triangle of this gram matrix represents the full statistics set. Synthesis proceeds by calculating loss on statistics between a target and noise seed image, backpropagating this loss through the network to adjust pixel values until a synthesized image is produced. Right: To learn a reduced statistics set, a single fully connected layer compresses to the Gram upper triangle to a reduced representation. These weights are trained using a contrastive loss, self-supervised by the objective of crops from the same texture image lying nearby in the reduced feature space.

Early approaches to texture synthesis are deeply rooted in efforts to model human vision both in neuroscience and computer vision. The field began with Julesz' (Julesz, 1962) work, which utilized

N-th order statistics in pixel space. Texture representation in Fourier space (Matsuyama et al., 1983), and later in pyramid space (Burt and Adelson, 1987; Freeman et al., 1991) improved texture models significantly (Heeger and Bergen, 1995). Along with considerable improvement in synthesis quality, came a strong link to the neuroscience of human visual processing, with the oriented multi-scale filters of the steerable pyramid Simoncelli and Freeman (1995) mirroring both the receptive fields of neurons found in area V1 (Turner, 1986; Malik and Perona, 1990), as well as their response properties (Olshausen and Field, 1997). The Portilla & Simoncelli model (Portilla and Simoncelli, 2000) further improved synthesis quality by including pixel, autocorrelation, and magnitude statistics on pyramid responses. For videos, models like Schödl et al. (2000) construct video texture by shuffling frames, but cannot model spatiotemporal texture.

These summary statistics texture models were later extended as peripheral vision models, with the incorporation of spatial pooling (Rosenholtz et al., 2012; Freeman and Simoncelli, 2011). This method involves synthesizing texture locally in many small overlapping regions, which when optimized jointly, lead to full-field image syntheses. By progressively increasing the size of these regions with increased distance from central vision, these models reflect loss of information in peripheral vision beyond the simple blurring of photoreceptor density loss. As analysis-by-sythesis models, they enable the synthesis of 'metamers' or 'mongrels'. These image syntheses appear lossy when viewed foveally (with central vision), but appear normal when viewed peripherally (metamers), and have been shown to capture the information available to human peripheral vision, therefore predicting human performance on a wide range of visual tasks (mongrels). These models have been since extended to incorporate color (Wallis et al., 2017), GPU acceleration (Brown et al., 2021), and expanded the field of view (Broderick et al., 2023). These models have also been combined with deep learning to explore higher-level similarities and divergences in human and machine vision (Harrington et al., 2024). Despite these advances, along with extensive knowledge of the spatiotemporal sensitivity parameters of human vision (Kelly, 1979; Krajancich et al., 2021) no work to our knowledge has yet been able to successfully describe the summary statistics set needed to extend these pyramid-based texture and peripheral vision synthesis models into the temporal domain.

A line of work in deep learning offers a way to generalize these statistics, enabling a path to biologically plausible spaiotemporal texture and peripheral syntheses. Style transfer (Gatys et al., 2016) is the extension of a spatial texture method (Gatys et al., 2015) which synthesizes textures by matching summary statistics defined as correlations between layers of pre-trained image networks such as VGG-Net (Simonyan and Zisserman, 2014). Like the classical models, this texture synthesis method has also been extended to peripheral vision to create successful syntheses (Deza et al., 2017; Wallis et al., 2017). On one hand, these methods demonstrate strong results in terms of synthesis quality while eliminating the need for hand-curated statsitics sets. However, compared to pyramid-based models they have three major disadvantages: 1) They are highly-overparametrized, with orders of magnitude more statistics. 2) They replace human visual neuroscience-informed filters with non-biologically-feasible deep networks. 3) Their summary statistics of deep network hidden layer correlations are difficult to interpret.

In addition to neural-inspired approaches, pure computer vision and deep learning approaches have had success in realistic synthesis of both static and dynamic textures. In the spatial domain, state of the art texture synthesis approaches often use diffusion models and generate photorealistic results (Chen et al., 2023; Youwang et al., 2024). For dynamic texture synthesis, early work (Doretto et al., 2003) laid the groundwork for representing the spatiotemporal dependencies needed in this task. Many modern spatiotemporal models build on the style transfer approach (Gatys et al., 2016). This same model has been shown to also succeed for single-layer networks (Ustyuzhaninov et al., 2017). The addition of long-range spatial and temporal dependencies have been shown to produce good spatiotemporal syntheses (Zhang et al., 2021), and the neural cellular automaton model has also been used to generate controllable texture synthesis in real time (Pajouheshgar et al., 2023).

## 3 SPATIAL MODEL

We design a simple model (Figure 1) that combines the interpretable and biologically-inspired pyramid filters of Portilla and Simoncelli (2000) with the powerful Grammian matrix representation of (Gatys et al., 2015). Our model leverages the GPU-accelerated multi-scale pyramids from (Brown et al., 2021), convolving input images with pyramid filters at individual combinations of orientation, scale, and color, the output we call 'pyramid images'. We treat each pyramid image as a channel,

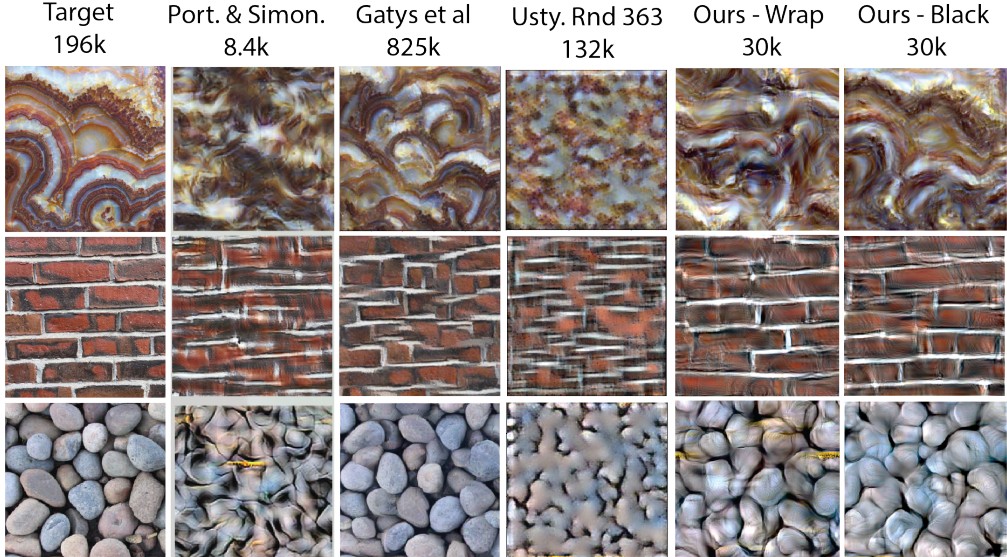

Figure 2: We generate texture syntheses with better quality than both the Portilla & Simoncelli and Ustyuzhaninov models. Our performance rivals Gatys et al, with an order of mangitude fewer parameters. P&S uses a similar multi-scale gabor strategy, but with hand-picked statistics. Gatys matches the Grammian of VGG-19. STGN uses the Grammian of mulit-scale gabors. STGN-Wrap use a wrapped Fourier transform, conserving purely spatially-invariant information, while STGN-Black uses zero-padding, and retains more spatial image content. Ustyuzhaninov et al uses a large set of random convolutional filters.

calculating the pair-wise correlations between each pyramid image pair, collapsing over space as a gram matrix. We use the upper triangle and diagonal of this matrix as the full statistics set for downstream analysis.

We apply our model in an analysis-by-synthesis framework to generate novel textures images, matching the statistics set of an input image, and synthesizing new textures by iterating noise images to match the statistics set. We show significant improvement over previous multi-scale-pyramid synthesis methods (Portilla and Simoncelli, 2000), and similar results to much larger, neural-network based synthesis methods (Gatys et al., 2015), despite over an order of magnitude smaller statisitcs set. Our method strongly outperforms the Grammian method on single-layer random filters (Ustyuzhaninov et al., 2017; Mongia et al., 2016) in both quality and model size (Figure 2).

An advantage of our method is correlations between known filters can be interpreted, and quality of syntheses attributed to different statistical families. We organize these families into two groups: one of statistics with one or more non-subband (non-pyramid level) filter image, and one group with correlations exclusively between subband images (Appendix Table 1). We organize the subband group into sub-groups, based on which properties are shared and differ between the pyramid images correlated.

The hand-curated statistics sets from previous models (Portilla and Simoncelli, 2000) rely heavily on correlation statistics from groups analogous to the first 6 subband families, which we call 'structured' statistics. These share all properties except for one (i.e. sub_Xlevel at neighboring levels). To attribute the quality of synthesis to different family groups, we perform an ablation experiment, synthesizing textures with only subsets of the statistical families (Appendix Figure 9). Interestingly, we find that these 'structured' correlation statistics generate poor synthesis even when combined together. By contrast, larger families of unstructured statistics generate superior syntheses, though we note these represent a larger portion of the statistics.

## 3.1 PARAMETER REDUCTION

To further investigate the contribution of individual texture families to synthesis quality, we use contrastive learning to reduce the correlation statistics to a compressed feature vector that can group textures (Figure 1). We crop texture images from (Cimpoi et al., 2014), and train a single fully connected layer using InfoNCE (Oord et al., 2018) to group texture crops from the same parent texture in the compressed space. We find that the trained single layer is sufficient to group same textures in embedding space (Appendix Figure 10). Here, we solve the contrastive learning problem as is done typically, using gradient-based optimization. We note however, that in this case with a single-layer encoder, it can also be solved analytically; we derive this solution (Appendix A.13).

Next, to evaluate the importance of different statistics to the contrastive learning task, we ordered statistics from most to least important based on both the absolute value of their weightings, as well as by their Shapley values (Roth, 1988) (not shown, similar results) (Figure 11). Again, we find correlations between subband filter outputs with few or no shared attributes are most important to the contrastive learning task. By contrast, correlations between lowpass filter outputs are least important for grouping textures. We visualize the relative contributions of individual families to the total statistics set when ordered by importance (Figure 12), and find that sub_Xmulti, highpass, sub-magnitude_Xmulti, and pass_multi families make an out-sized contribution to the most important statistics.

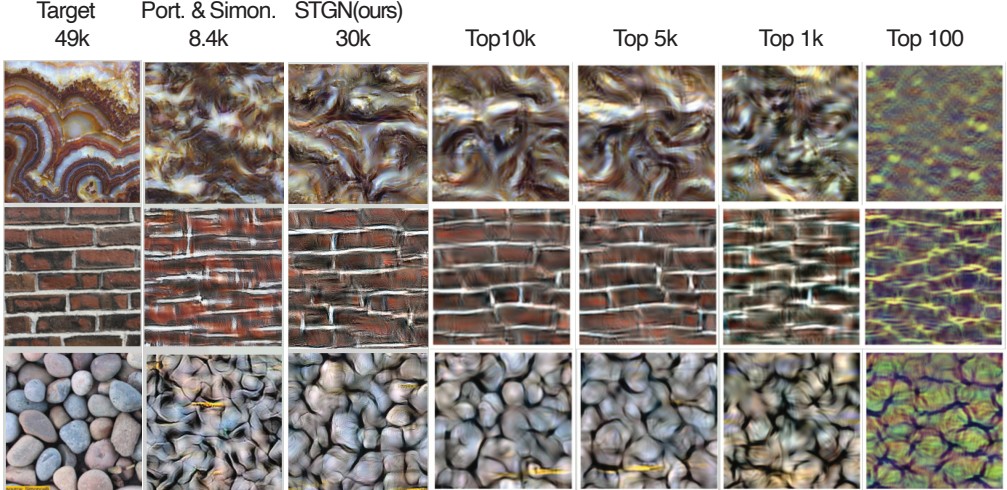

Figure 3: Syntheses optimized with depleted statistics sets, chosen by most to least important on the contrastive learning task.

We evaluate the transfer of importance in the contrastive learning to synthesis quality, again performing an ablation experiment, but optimizing using only statistics of the highest weighting in the contrastive learning task (Figure 3). We find that indeed, importance in contrastive learning directly impacts with synthesis quality. In fact, by choosing the the most important statistics, we are able to achieve good syntheses that outperform the Portilla & Simoncelli model with only 5,000 of the total 28,929 statistics, for a reduction of 83%. By contrast, in a control experiment, we find that the statistic groups of the same size from the least important statistics set are extremely poor, and randomly selected are only slightly better (Figure 14).

We qualitatively evaluate of our ablation syntheses, demonstrating the effect on a subset of the DTD validation set (Figure **??**). Matching qualitative visual analysis (Figure 3), we find that the quality for our texture syntheses is generally similar until the number of stats drops below 1000. Perceptual loss metrics (MS SSIM, FSIM, LPIPS) show the best performance for the top-N statistics as compared to random and bottom ranked size-matched sets. By contrast, non-perceptual losses have non-intuitive results. FID and KID Mean report similar performance for the size-matched random set and top sets, whereas MSE reports the *worst* performance with the top-N statistics. This does not reflect

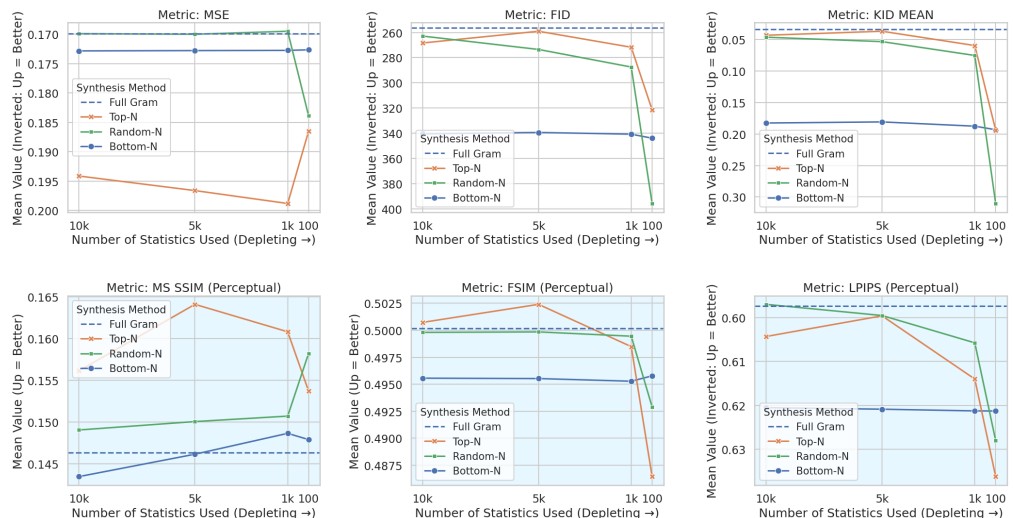

Figure 4: Results for quantitative comparison of texture image synthsis quality for depletions for a 3-per-class subset of DTD-val1. Metrics MSE, FID, KID MEAN, MS SSIM, FSIM, LPIPS. Perceptual losses highlighted in blue.

the perceptual quality visually (Figure 14), but is somewhat unsurprising given our statistics enforce statsitical correlations broadly, not pixel-level values. .

## 3.2 Peripheral Synthesis

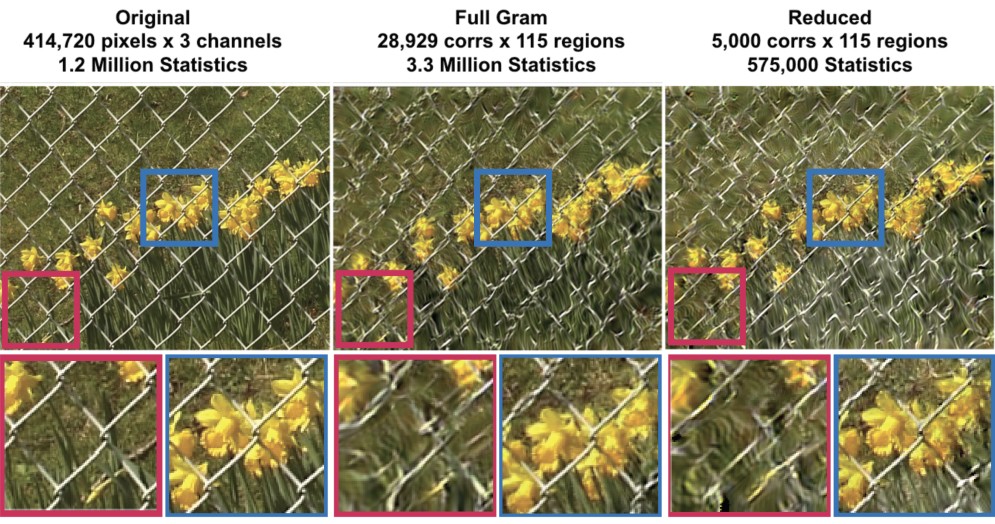

Figure 5: Synthesis incorporating spatial pooling. Spatial pooling of textures creates synthesis useful for studying human peripheral vision, where the center of the visual field is rendered at high fidelity, and the edges of vision are lossy. Original image (left) represents 1.2 Million values per image. Syntheses using the full model (middle) with 28,920 statistics per pooling region, and 115 pooling regions increases the representation size to 3.3 million, and creates quality syntheses. Our reduced model (right) decreases the statistical representation to 5,000 statistics per pooling region, for 575,000 total values, while still achieving good results. This results in a greater than 50% compression compared to the input image.

To demonstrate the utility of our model for studying human peripheral vision, we combine our statistical synthesis method for textures with spatial pooling as in (Freeman and Simoncelli, 2011;

Rosenholtz et al., 2012), generating foveated synthesis on full-size images which have non-uniform texture (Figure 5). Our model smoothly captures the transition from pixel-perfect at gaze center, to the strong spatial distortions of metamers and mongrels, a testament to the lossy encoding of human peripheral vision. Utilizing the reduced representation learned from contrastive learning, we are able to reduce the size of our peripheral syntheses significantly, while retaining high quality. Future work will evaluate these peripheral syntheses with our reduced statistics set on human subjects.

## 4 SPATIOTEMPORAL MODEL

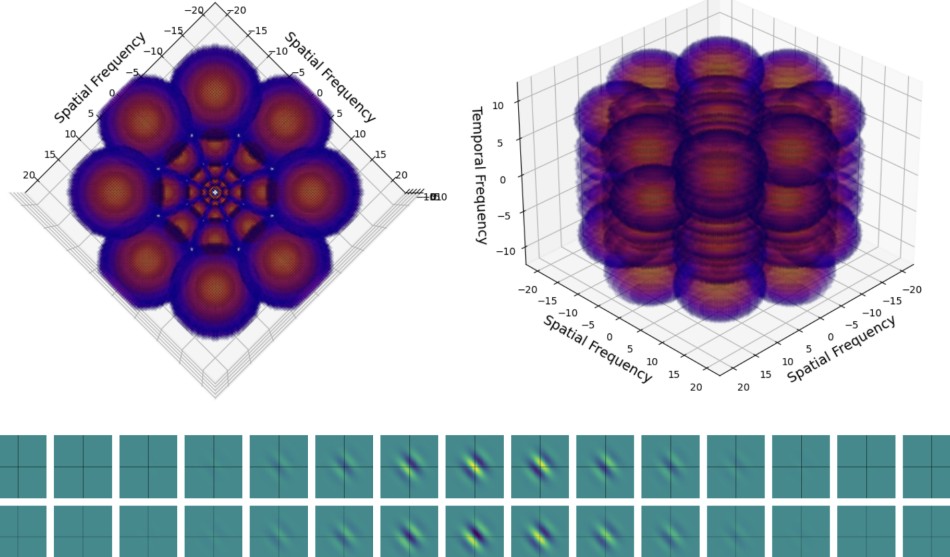

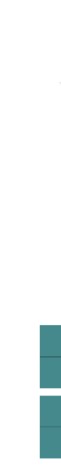

Figure 6: We design a filter bank of spatiotemporal Gabor filters that tile the spatiotemporal frequency space. Viewed in Fourier space (top), filters are arranged in concentric rings that are stacked vertically, selecting for 8 orientations at increasing spatial and temporal frequencies. Each filter in the positive temporal frequency space has a quadrature pair mirrored across a line through the origin. Left: top view, Right: side view. Bottom: A real+imaginary pair of spatiotemporal filters, generated by inverse Fourier transform of the spatiotemporal frequency defined filters above. Left to right denotes the temporal axis, with frames sub-sampled for visualization. This pair of filters is sensitive to low spatial and low temporal frequency motion to the upper-right.

Extending the Portilla–Simoncelli image model to the spatiotemporal domain is impractical because its hand-selected cross-scale and cross-orientation statistics do not generalize across the time domain. Instead of hand picking important statistics, our method, leveraging all combinations of filter correlation statistics with the gram matrix, can be naturally extended to video textures. The first step for expansion of our model into the temporal domain is the design of a human-vision informed spatiotemporal filter bank. We extended the classical multi-scale spatial pyramid (Heeger and Bergen, 1995) into the time domain with a set of spatiotemporal filters centered around the spatiotemporal frequency space of human visual sensitivity (see Appendix). A subset of these filters in the Fourier domain with boundaries at half maximum amplitude is shown in Figure 6 (Top). See A.10 for the mathematical parameterization for these spatiotemporal filters. When the inverse Fourier transform of a quadrature pair of frequency defined filters is taken, both a real and imaginary component is generated. These spatiotemporal filter pairs can be visualized in the pixel-frame space, enabling intuition for the type of motion a filter pair selects for. A sample of these filter pairs is shown in Figure 6, bottom.

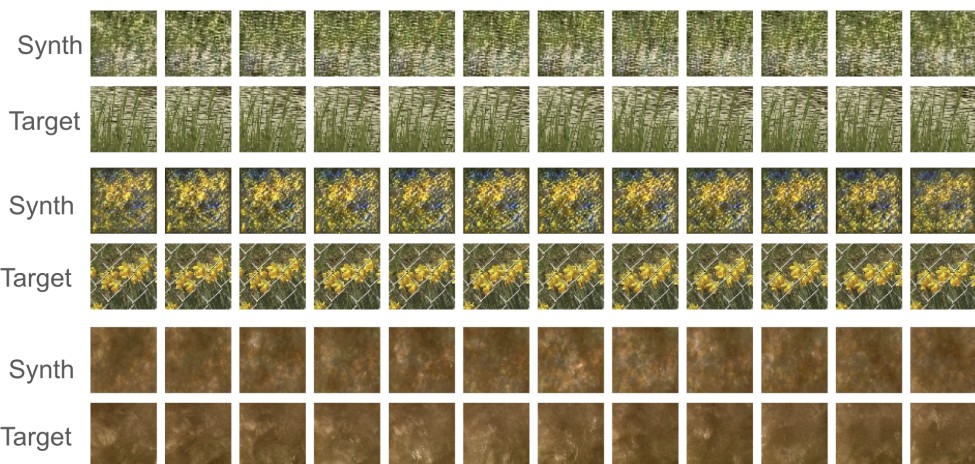

Figure 7: Our spatiotemporal model produces video syntheses that match well the appearance of the input video. Examples demonstrate the wide range of texture videos well-captured by our model. 'Target' indicates the original video, and 'Synth' indicates the video synthesis generated by our model by matching gram matrix statistics. Note that syntheses do not reproduce the video exactly, but instead samples from the space of all videos with the same statistical distribution.

### 4.1 SYNTHESIZED TEXTURE VIDEOS

We apply the same gram matrix model to calculate correlations between our spatiotemporal filters, synthesizing texture videos by matching the set of filter correlation statistics. While contrastive learning in the spatiotemporal case is not computationally feasible given resource constraints, we are able to reduce our model size somewhat based on our findings from the spatial-only case, removing lowpass filters. Our model synthesizes videos that visually match well to the target textures (Figure 7), and work well over a set of textures videos chosen to maximize variety in structure and motion properties, while maintaining computational tractability (Table 3). Importantly, our syntheses retain recognizable temporal structure, which frame-wise spatial synthesis cannot reproduce. Unlike the spatial case, we find that for video textures, histogram matching over the whole image is needed in addition to statistics matching to achieve good syntheses. To our knowledge, no prior perceptually grounded, human-vision–based texture model has been realized for video.

## 5 DIFFUSION MODELS

Diffusion models are fast and efficient methods to sample high quality images from the image manifold, as opposed to pixel-based backpropagation which is relatively slow, and can produce non-photorealistic syntheses. We test the ability of our texture statistics to guide diffusion synthesis in both pixel-space and latent diffusion models, testing diffusion models with varying architectures, complexity levels, and schedulers.

We find that gradient-guided diffusion using our statistics set is able to move synthesis towards the target texture, evidenced both perceptually and by a significant reduction in the statistics loss (Figure 8). For gradient guidance with simple optimization schemes, diffusion models enforce syntheses to be closer to natural images, with a strong bias for objects, due to this bias in the training data for most diffusion models. This lead to results reminiscent of style transfer (Gatys et al., 2016). For these vanilla optimization schemes, synthesis was extremely sensitive to optimizer/step-size tuning, was successful only for pixel-based diffusion (DDPM), and struggled to achieve the quality or magnitude of statistical match compared to pixel backpropagation. Through this naive method, we noted that as guidance strength increased, before the magnitude became sufficient to enforce texture statistics, the diffusion process invariably overshot the image manifold (See Appendix A.14).

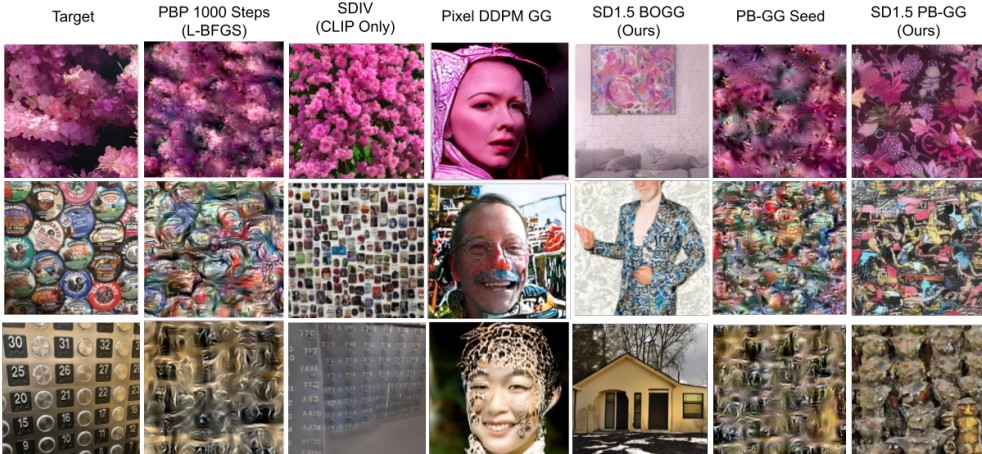

Figure 8: Variants of gradient guidance procedure on diffusion models impart texture of the target image to various degrees, while remaining on the naturalistic image manifold. Simpler guidance procedures achieve a style-transfer like effect, while more complex optimization and seeding schemes can escape the diffusion models object bias and achieve texture images with similar properties to the target. While these images differ significantly both semantically and visually when foveated, we invite the reader to test these textures as peripheral mongrels/metamers by directing your gaze to the center column of face images. Can you tell the difference between the target texture (far left) and the PB-GG (far right) images with your peripheral vision?

To address this, we developed two custom optimization schemes. The first, back-off gradient guidance (BOGG), runs vanilla gradient guidance with N warm restarts, extending the effective gradient guidance process without extending diffusion denoising. This method enabled similar style transfer-like results for latent diffusion models including SD1.5. Our second method, pixel backprop-gradient guidance (PB-GG), runs pixel backprop (L-BFGS) for 25 iterations, then adds an appropriate level of noise to initialize the diffusion process at an intermediate timestep, from which, the gradient guidance diffusion is then started. This method gives good quality results, which are photorealistic, are matched in style to the target texture and minimizes object bias.

Future work in leveraging diffusion models as optimizers for human-grounded texture synthesis could train a texture-specific diffusion model with a learned or statistic-constrained latent. This would reduce the object/shape-bias of models tested here, and improve stability compared to gradient guidance. Guiding video diffusion models on our spatiotemporal statistics would be fruitful.

# 6 DISCUSSION

We introduce GramStatTexNet, a texture synthesis model that merges the interpretability of biologically inspired multi-scale pyramid filters with the effectiveness of correlation-based methods. Our model achieves high-quality texture and peripheral vision syntheses that are on-par with the quality of deep learning approaches but with improved transparency and better neural-plausibility. By categorizing our statistics into families and utilizing contrastive learning, we highlight the benefits of this interpretable approach, and enable a huge reduction in model size, which we use to demonstrate the synthesis of metamers/mongrels using gaze-based spatial-pooling. In addition, the systematic nature of the Grammian in calculating filter correlations removes the need for a heuristically-chosen set of statistics - we leverage this to extended our method to the spatiotemporal domain, synthesizing texture videos. Finally, we sucessfully leverage diffusion models as optimizers for our statistics, generating textures that both lie on the natural image manifold, and match textures in a perceptually-informed manner. Our work offers a comprehensive framework that aligns with human visual perception across space and time, contributing to advancements in visual neuroscience, computational modeling, and computer graphics.

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

# A  APPENDIX

## A.1  LLM USE DISCLOSURE

We made limited use of large language models (LLMs) in the preparation of this work. LLMs were consulted to aid our understanding of gradient guidance methods and to identify relevant references during the early stages of research. In addition, LLMs assisted with coding: some functions were drafted directly by LLMs under our supervision, while we wrote and reviewed substantial portions of the code ourselves. Finally, LLMs were used to improve the clarity and readability of the manuscript. LLMs did not contribute to the generation of research ideas, experimental design, or results; their role was restricted to auxiliary support in learning, reference gathering, coding, and writing refinement.

## A.2  METHODS

### A.2.1  IMAGE AND VIDEO SYNTHESIS

We synthesized visualized textures following the synthesis procedure in (Koevesdi et al., 2023), using the filter portion of the codebase from (Brown et al., 2021), with the pyramid settings at 4 orientations, 6 edge levels, edge start at level 1, with marginal statistics included. For image synthesis on individual textures, we used the LBFG-S solver (Liu and Nocedal, 1989) as demonstrated for textures in Gatys et al. (2015). We found that this method generated very good quality synthesis. For quantitative results computed at scale, we found that the Hessian calculation in LBFG-S caused problems in independent batching of texture images, and was prohibitively slow espeically without the ability to batch. To solve this, we investigated alternative solvers and found that Madgrad (Defazio and Jelassi, 2021) gave comparable results to L-BFGS (Appendix Figure 13), but was fast and most importantly compatible with batching. We used this solver for synthesizing at scale and for quantitative results.

Syntheses for random filters (Ustyuzhaninov et al., 2017) were created by a custom Pytorch port from the official Theano repo (https://github.com/ivust/random-texture-synthesis) with 10k iterations and the Madgrad solver (Defazio and Jelassi, 2021). Synths for Gatys et al (Gatys et al., 2015) were created from unoffical Pytorch repo for (Gatys et al., 2015) (https://github.com/trsvchn/deep-textures). Synths for Portilla & Simoncelli were created using the official color extension package in Matlab (https://www.cns.nyu.edu/pub/eero/colorTextureSynth.tgz).

Spatial pooling of our model for peripheral syntheses was incorporated by tiling the image with overlapping pooling regions of 16x16 pixels tiled across the image, then synthesizing our texture model at each region in parallel. We utilize the pooling method from Brown et al. (2021), warping the image, synthesizing over uniformly-tiled regions in the warped image space, and un-warping after synthesis, with parameters matched as closely as possible to (Freeman and Simoncelli, 2011).

To synthesize novel videos matching the statistics of input videos, we follow the same procedure as for images, calculating the (video) statistics vector from a target video. Then, starting with a white-noise video, we use LBFG-S to adjust pixel values in the synthetic video, using backpropogation of the mean squared error in statistics space between the synthetic and target video. This results in a synthetic video that matches the statistics of the original target video, but with scrambling of absolute spatial and temporal localization. Given this loss of spatial and temporal information, we use texture videos from the DynTex Database (Péteri et al., 2010), which we convert to greyscale, spatially center-crop and downsample to minimize artifacts and ensure uniformity.

### A.2.2  SPATIOTEMPORAL FILTERS

We choose the range of the spatiotemporal space of up to 16 CPD and 20 Hz, following the human spatiotemporal contrast sensitivity calculations for foveal vision (Kelly, 1979), and spatiotemporal flicker fusion for peripheral vision (Krajancich et al., 2021). Within this range, we design filters as 3D Gaussians arranged in concentric stacked rings (shaped like a mochi-donut), similar to (Simoncelli, 1993). A single ring represents 8 filters of varying orientations (motion directions) at the same spatial and temporal frequency. Rings are arranged concentrically to select for increasing spatial frequencies, and stacked along the temporal axis to select for increasing temporal frequencies. Each

filter located in the positive half of the temporal frequency space has a quadrature pair filter mirrored in the negative half of the temporal frequency space, in the opposite spatial quadrant. To tile the space, we chose a set of 8 orientations, 4 spatial scales, and 3 temporal scales, with a real and imaginary pair each. We also include one spatial and one temporal low-pass filter, each with a real and imaginary pair, for a total of 200 filters. These spatiotemporal filter pairs correspond to a set of Gabor filters.

### A.2.3 DIFFUSION SYNTHESIS

For pixel diffusion, we utilize a DDPM posterior sampling method designed to guide diffusion in non-linear inverse regimes (Chung et al., 2022). For latent diffusion, we test both Stable Diffusion v1.5 Rombach et al. (2022) (DDPM) and Stable Diffusion v3 (Esser et al., 2024) (Rectified Flow Matching). For all models tested, we set text prompt guidance to zero, guiding from the Mean Square Error (MSE) loss for the full set of 28,929 texture statistics as calculated on the target texture and the Tweedie denoiser's (Efron, 2011) predicted image.

### A.2.4 DATASETS

For the spatial texture model, we utilized the Describable Textures Database (DTD) (Cimpoi et al., 2014), keeping the largest centered square and resizing the image to 256x256 for synthesis, and 5 smaller crops for the contrastive learning. For training the contrastive learning module, we used the entire training subset of split 1, which samples all 40 texture types. For the quantitative evaluations of our syntheses and depletions, for computational tractability, we sampled 3 randomly chosen images from each category of the validation subset of validation split 1, for a total of 141 images. For computational reasons, we do not evaluate (Portilla and Simoncelli, 2000), (Gatys et al., 2015), or (Ustyuzhaninov et al., 2017) on this full validation set, but show them for qualitative comparison on demo images (Figure 3).

For the video texture model, we utilized the Dynamic Texture Database (DynTex) Péteri et al. (2010). Again, for computational tractability, we synthesized a subset of videos selected to vary in frequency, shape, and motion type.

### A.2.5 CONTRASTIVE LEARNING

For contrastive learning, we trained a single fully connected linear layer of size (28,929 x 100), reducing the size of our representation space by nearly 300 times. We used DTD train split 1, taking 5-crops (4 corners plus center) of the image of size 128x128 with random vertical and horizontal flips, and feeding these through the analysis pipeline, extracting the upper triangle of the Grammian matrix for 28,929 statistics, and training the network in batches of 50 textures (200 crops total). We trained the network for 100 epochs with the Normalized Temperature-Scaled Cross-Entropy (NTXendt) loss from pytorch metric learning. Conceptually, this loss encourages the learned representation (100 dimensions) to represent textures from the same parent texture similarly, and textures from different parent textures differently.

### A.3 ABLATION FAMILIES

### A.4 CONTRASTIVE LEARNING

### A.5 CONTRASTIVE-LEARNED WEIGHT INTERPRETATION

### A.6 CONTRIBUTION OF STATISTICAL FAMILIES

#### A.6.1 OPTIMIZER: L-BFGS VS MADGRAD

L-BFGS is often the optimizer of choice for image synthesis due to it's speed and efficiency at jointly optimizing the pixel space to find the target image, due to it being a second order method. However, the calculation of the Hessian makes batching image syntheses problematic, as gradients are difficult to keep separate from separate images, and can make large scale synthesis intractable. We explored the use of L-BFGS as an alternative optimizer that is batch-compatible, and produces synthesis comparable to L-BFGS, the de-facto gold standard for pixel-backpropogation based synthesis.

Figure 9: Depleted syntheses optimized using statistics subsets defined by categorical families. Families with 'structured' statistics pairs where only one parameter differs generate poor synthesis compared to larger families of unstructured pairs.

## A.7 DEPLETED SYNTHESES CONTROL

## A.8 QUANTITATIVE METRICS FOR DEPLETED SYNTHESES

## A.9 SPATIAL PYRAMID STATISTICS FAMILIES

## A.10 SPATIOTEMPORAL FILTER EQUATIONS

In the Fourier domain, our filter banks take the form of anisometric Gaussian functions centered on $(\mu_x, \mu_y, \mu_t)$ with variances $(\sigma_x^2, \sigma_y^2, \sigma_t^2)$

$$\tilde{f}(\mathbf{k}) \propto \exp\left[ -\frac{1}{2}\left( \left(\frac{k_x - \mu_x}{\sigma_x}\right)^2 + \left(\frac{k_y - \mu_y}{\sigma_y}\right)^2 + \left(\frac{k_t - \mu_t}{\sigma_t}\right)^2 \right) \right].$$

$(1)$

The filter positions are determined by the spatial scale $(\rho)$, temporal scale $(\zeta)$, and orientation $(\phi)$. These are defined in cylindrical coordinates as:

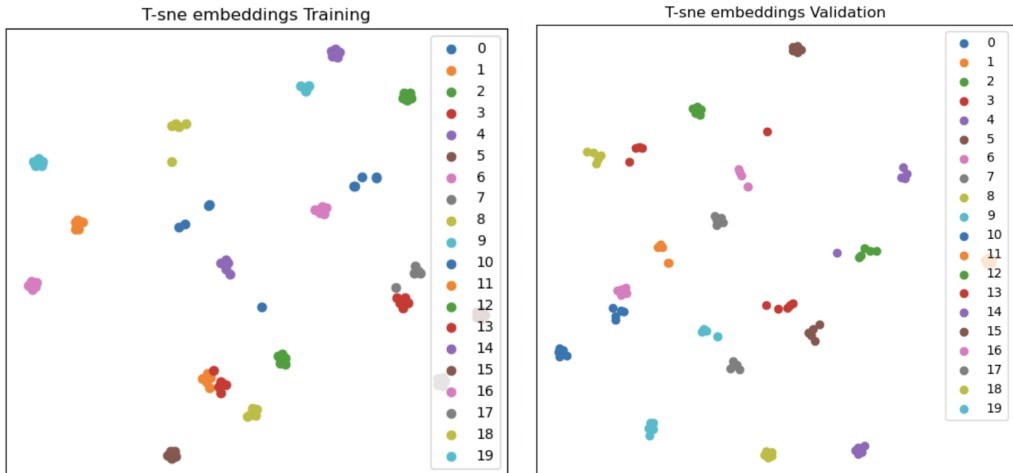

Figure 10: Contrastive learning successfully groups texture images cropped from the same parent image (same color), and separates images from different parent images. Training set (left). Validation set (right)

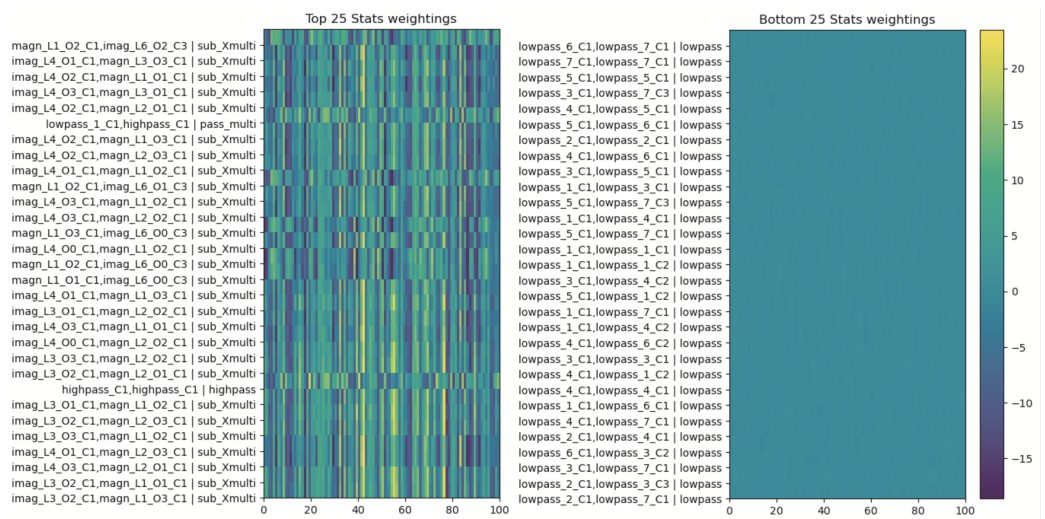

Figure 11: We use the weight matrix of the compression layer from the contrastive learning module to order correlation statistics by their contribution to the compressed representation. Learned weights of the single fully connected layer show high weightings for unstructured statistics, and lowest weightings for lowpass filter statistics.

$$\mu_x = \rho \cdot \cos(\phi) \tag{2}$$
$$\mu_y = \rho \cdot \sin(\phi) \tag{3}$$
$$\mu_t = \zeta \tag{4}$$

Finally, the standard deviations $\sigma_i$ scale with the centers via spatial and temporal scaling factors $\alpha_S = 0.4$ and $\alpha_T = 0.7$ such that:

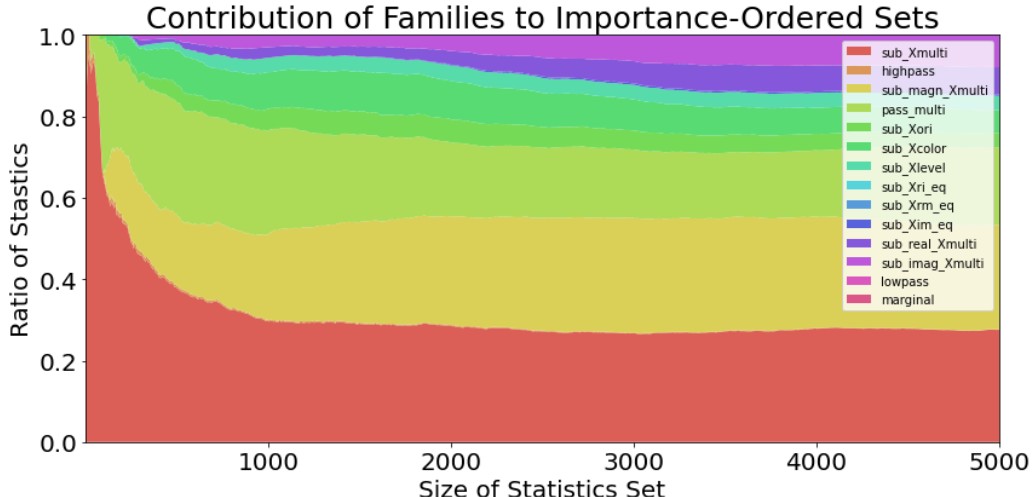

Figure 12: Using the contrastively learned importance ordering, we visualize the percent contribution of individual families to the total set, for varying statistics set size. Families sub_Xmulti, highpass, and pass_multi contribute most when selecting for the most important statistics (left).

| Non-Subband Pyramid Stats | | | |
|---|---|---|---|
| **Group Name** | **Stat A** | **Stat B** | **Number Stats** |
| marginal | mean — var — std | mean — var — std | 9 |
| highpass | highpass | highpass | 6 |
| lowpass | lowpass | lowpass | 231 |
| pass_multi | highpass — lowpass | X | 5247 |
| **Subband Pyramid Stats** | | | |
| **Group Name** | **Stats Same** | **Stats Differ** | **Stats** |
| sub_Xori | Level, Color, Pyr | Ori | 324 |
| sub_Xcolor | Level, Ori, Pyr | Color | 432 |
| sub_Xlevel | Color, Ori, Pyr | Level | 540 |
| sub_Xri_eq | Level, Color, Ori | RealXImag | 72 |
| sub_Xrm_eq | Level, Color, Ori | RealXMag | 72 |
| sub_Xim_eq | Level, Color, Ori | ImagXMag | 72 |
| sub_real_Xmulti | Real | Level, Color, Ori | 2196 |
| sub_imag_Xmulti | Imag | Level, Color, Ori | 2196 |
| sub_magn_Xmulti | Magn | Level, Color, Ori | 2196 |
| sub_Xmulti | - | Pyr, Level, Color, Ori | 15336 |

Table 1: Summary of statistical families for spatial model. Non-subband statistics correlate at least one non-subband pyramid statistic. Subband pyramid statisitcs are exclusively between pyramid subbdand (edge) images.

$$\sigma_x = \alpha_S \cdot \mu_x \tag{5}$$
$$\sigma_y = \alpha_S \cdot \mu_y \tag{6}$$
$$\sigma_t = \alpha_T \cdot \mu_t \tag{7}$$

For our filter bank, we sample the spatial scales, temporal scales, and orientations over the ranges:

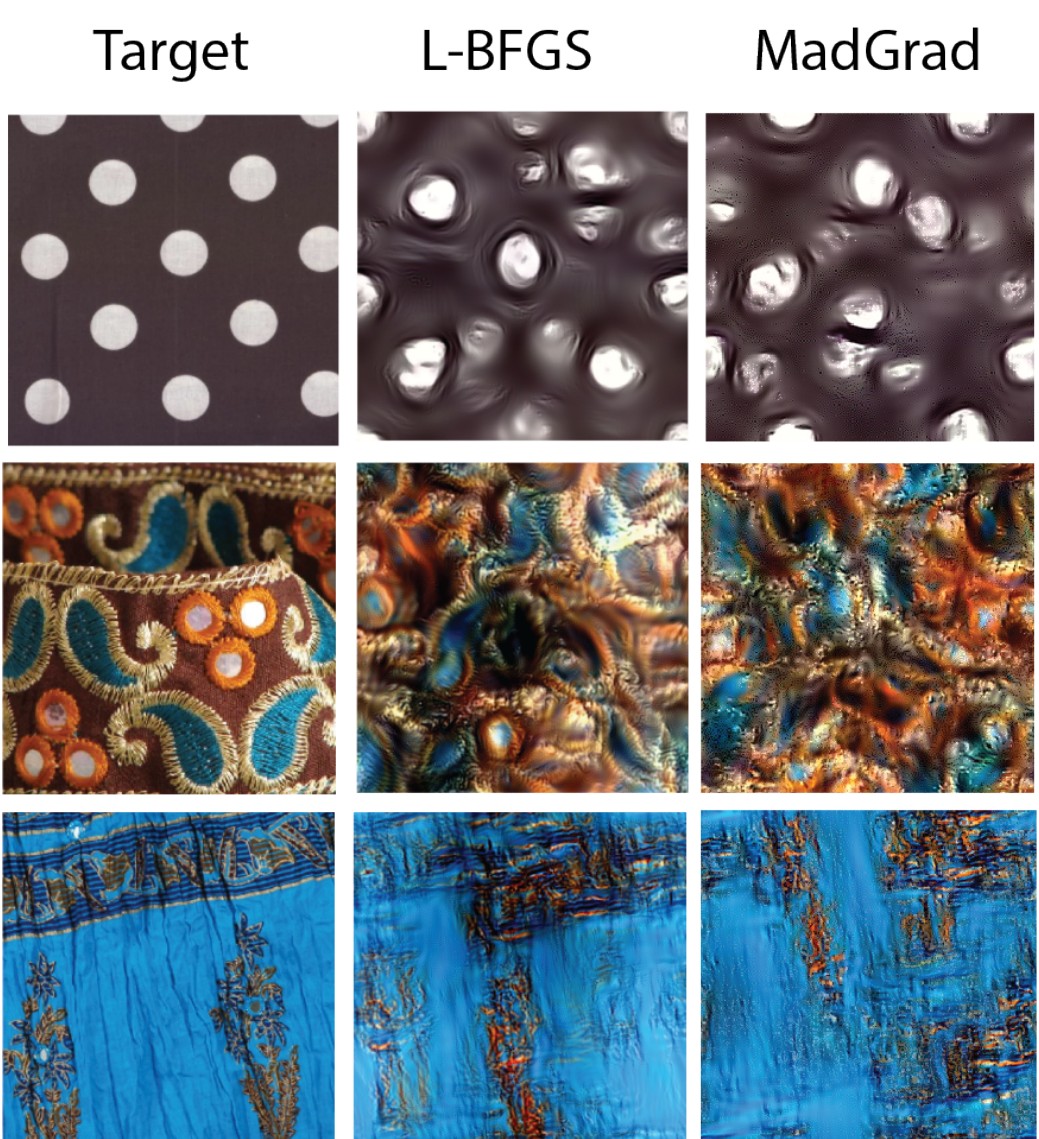

Figure 13: We find that Madgrad produces good quality synthesis, and is batch compatible, saving significant time when synthesizing at scale.

$$\rho \in \{1, 2, 4, 8, 16\} \tag{8}$$

$$\zeta \in \{1, 2, 4, 8, 16\} \tag{9}$$

$$\phi \in \{0, \frac{\pi}{4}, \frac{\pi}{2}, \frac{3\pi}{4}, \pi, \frac{5\pi}{4}, \frac{3\pi}{2}, \frac{7\pi}{4}\} \tag{10}$$

### A.11 SPATIOTEMPORAL STATISTICS FAMILIES

A benefit of our approach is that as a gram matrix element, each statistic represents a correlation between pairs of known filter responses, making the set of statistics easily interpretable. This is true for both the spatial-only filters as well as the spatiotemporal filters. For the spatiotemproal filters, given the much larger number of filters, we use a slightly different organizational approach, separating them by each parameter that varies. Table 2 organizes these statistics into families based on the relationship between the two filter images being correlated. We take specific note of the

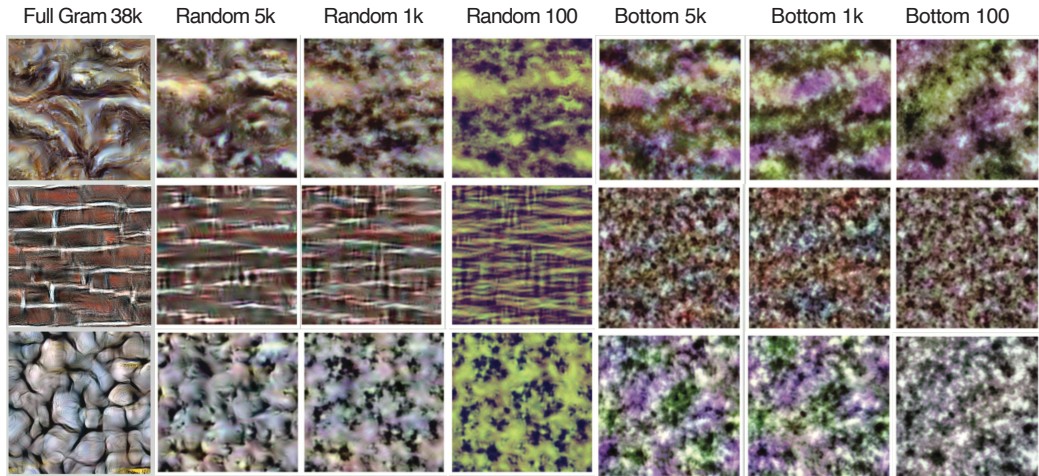

Figure 14: Control experiments with syntheses optimized with depleted statistics sets, using the least important and randomly chosen statistics. These synthesis perform extremely poorly, indicating that contrastive importance order correlates directly with synthesis quality.

groups denoted by a single letter (C,O,S,T,P), as these correspond to correlations between filters that vary in only one attribute. These statistics are analogous to the hand-selected spatial statistics in previous pyramid-based texture and peripheral vision models including (Portilla and Simoncelli, 2000; Freeman and Simoncelli, 2011; Balas, 2006). In addition, spatial texture investigations have shown that low-pass filter correlation statistics (LPS) are less important to successful synthesis, and correlation statistics across many filter attributes (such as COSTP) are most important (DuTell et al., 2023).

## A.12 VIDEO SYNTHESES

Videos of our model syntheses can be accessed for double blind review at: https://drive.google.com/drive/folders/1H7mM51UGErmCRGu0HmIsZeGhYaOYRHqL

| AUTO | Auto-correlation | 1212 |
|---|---|---|
| LPS | Spatial Low-pass Corr. | 7233 |
| LPT | Temporal Low-pass Corr. | 7233 |
| C | Cross: color. Same: orientation, spatial, temporal, phase | 1200 |
| O | Cross: orientation. Same: color, spatial, temporal, phase | 4200 |
| S | Cross: spatial. Same: color, orientation, temporal, phase | 2400 |
| T | Cross: temporal. Same: color, orientation, spatial, phase | 2400 |
| P | Cross: phase. Same: color, orientation, spatial, temporal | 600 |
| CO | Cross: color, orientation. Same: spatial, temporal, phase | 8400 |
| CS | Cross: color, spatial. Same: orientation, temporal, phase | 4800 |
| CT | Cross: color, temporal. Same: orientation, spatial, phase | 4800 |
| CP | Cross: color, phase. Same: orientation, spatial, temporal | 1200 |
| OS | Cross: orientation, spatial. Same: color, temporal, phase | 16800 |
| OT | Cross: orientation, temporal. Same: color, spatial, phase | 16800 |
| OP | Cross: orientation, phase. Same: color, spatial, temporal | 4200 |
| ST | Cross: spatial, temporal. Same: color, orientation, phase | 9600 |
| SP | Cross: spatial, phase. Same: color, orientation, temporal | 2400 |
| TP | Cross: temporal, phase. Same: color, orientation, spatial | 2400 |
| COS | Cross: color, orientation, spatial. Same: temporal, phase | 33600 |
| COT | Cross: color, orientation, temporal. Same: spatial, phase | 33600 |
| COP | Cross: color, orientation, phase. Same: spatial, temporal | 8400 |
| CST | Cross: color, spatial, temporal. Same: orientation, phase | 19200 |
| CSP | Cross: color, spatial, phase. Same: orientation, temporal | 4800 |
| CTP | Cross: color, temporal, phase. Same: orientation, spatial | 4800 |
| OST | Cross: orientation, spatial, temporal. Same: color, phase | 67200 |
| OSP | Cross: orientation, spatial, phase. Same: color, temporal | 16800 |
| OTP | Cross: orientation, temporal, phase. Same: color, spatial | 16800 |
| STP | Cross: spatial, temporal, phase. Same: color, orientation | 9600 |
| COST | Cross: color, orientation, spatial, temporal. Same: phase | 134400 |
| COSP | Cross: color, orientation, spatial, phase. Same: temporal | 33600 |
| COTP | Cross: color, orientation, temporal, phase. Same: spatial | 33600 |
| CSTP | Cross: color, spatial, temporal, phase. Same: orientation | 19200 |
| OSTP | Cross: orientation, spatial, temporal, phase. Same: color | 67200 |
| COSTP | Cross: color, orientation, spatial, temporal, phase. Same: | 134400 |
| TOTAL | All Filters | 735078 |

Table 2: We group the 735078 spatiotemporal correlation statistics into families based on the filter attributes that are cross-correlated. These attributes are: color channel, motion direction (orientation), spatial frequency selectivity (scale), temporal frequency selectivity (scale), and real/imaginary (phase), as well as spatial lowpass and temporal lowpass filters. The 'AUTO' family contains the filter autocorrelations, located on the diagonal of the gram matrix. 'COSTP' group contains all the statistics between filters that vary in all of the attributes.

| Video | LP_ALX | LP_VGG | SSIM | FID | MSE | PSNR |
|-------|--------|--------|------|-----|-----|------|
| Grass | 0.1924 | 0.4243 | 0.9492 | 81.75 | 0.0305 | 15.16 |
| Fence | 0.3615 | 0.6073 | 0.9882 | 372.00 | 0.0618 | 12.10 |
| Fire | 0.1954 | 0.3837 | 0.7712 | 72.46 | 0.0159 | 18.10 |

Table 3: Quantitative comparison of synthesis quality for movies tested. Synthesis Metrics: LPIPS (AlexNet and VGGNet), SSIM, FID, MSE, and PSNR (DB).

### A.13 ANALYTICAL SOLUTION FOR CONTRASTIVE LEARNING

Let $i$ denote the $i$th parent texture images, and $n(i)$ denotes the set of texture images cropped from $i$th parent texture images. Let $a(x)$ denote the texture statistic of image $x$, and let $P$ be the weight of the fully connected layer of the contrastive learning. The contrastive learning problem can be formulated as:

$$\min_P \sum_i \sum_{j,k \in n(i)} ||Pa(x_j) - Pa(x_k)||_F^2 \tag{11}$$

such that

$$P\mathbb{E}_x[a(x)a(x)^T]P^T = I \tag{12}$$

This optimization problem can be written in the following matrix form:

$$\min_P ||PAD||_F^2 \quad s.t. \quad PVP^T = I \tag{13}$$

Where $V = \frac{1}{N}$, N is the total number of used images crops. $V$ is the covariance matrix of statistics. $A$ contains the statistic of all possible crops. Each row of $A$ is the $a(x_i)$ for some $i$. $D$ is analogous to the first order derivative operator defined on a graph. Specifically, $D = [d_1, d_2, \cdots, d_M]$, where $M$ denotes the number of pair of crops in the training set. For $i$th pair of crops $x_k$ and $x_j$, we have $d_{ik} = 1$ and $d_{ij} = -1$. All other entries of $D$ are zeros.

The solution to this generalized eigen-decomposition problem is given by $P = UV^{-\frac{1}{2}}$, where $U$ is a matrix of $L$ trailing eigenvectors (i.e. eigenvectors with the smallest eigenvalues) of the matrix $Q = V^{-\frac{1}{2}}ADD^T A^T V^{-\frac{1}{2}}$ (Chen et al., 2022).

This means solving the contrastive learning problem requires calculating $Q$, which requires calculating the matrix $ADD^T A^T$ and $V^{-\frac{1}{2}}$. The latter can be easily computed by diagonalizing $A$. To compute $ADD^T A^T$ on the other hand is generally computational expensive. Luckily, since $D$ is highly sparse, we can efficiently calculate matrix $ADD^T A^T$ as the following:

$$Q = 2\left(\sum_i |n(i)| \sum_{k \in n(i)} a(x_k)a(x_k)^T - \bar{a}_i \bar{a}_i^T\right) \tag{14}$$

where $\bar{a}_i = \sum_{j \in n(i)} a(x_j)$. Let $a_i$ denote $a(x_i)$ for convenience. The derivation is the following:

$$Q = ADD^T A^T$$
$$= \sum_i \sum_{j,k \in n(i)} (a_k - a_j)(a_k - a_j)^T$$
$$= \sum_i \sum_{j \in n(i)} \sum_{k \in n(i)} a_k a_k^T + a_j a_j^T - a_k a_j^T - a_j a_k^T$$
$$= \sum_i \sum_{j \in n(i)} ( \sum_{k \in n(i)} a_k a_k^T + |n(i)| a_j a_j^T -$$
$$( \sum_{k \in n(i)} a_k )a_j^T - a_j ( \sum_{k \in n(i)} a_k )^T )$$
$$= \sum_i (|n(i)| \sum_{k \in n(i)} a_k a_k^T + |n(i)| \sum_{j \in n(i)} a_j a_j^T -$$
$$( \sum_{k \in n(i)} a_k )( \sum_{j \in n(i)} a_j )^T -$$
$$( \sum_{j \in n(i)} a_j )( \sum_{k \in n(i)} a_k )^T )$$
$$= \sum_i 2|n(i)| \sum_{k \in n(i)} a_k a_k^T - 2( \sum_{k \in n(i)} a_k )( \sum_{j \in n(i)} a_j )^T$$
$$= 2( \sum_i |n(i)| \sum_{k \in n(i)} a_k a_k^T - \bar{a}_i \bar{a}_i^T )$$

### A.14 TEXTURE GENERATION WITH GRADIENT GUIDED DIFFUSION MODELS

As described in (Sec. 5), we employed diffusion models for more photorealistic generation of images that were constrained by our texture statistics model.

In the unconditioned Denoising Diffusion Probabilistic Model, the inverse denoising steps can be described by the following stochastic differential equation:

$$d\mathbf{x}_t = \left[ -\frac{\beta(t)}{2}\mathbf{x}_t - \beta(t)\nabla_{\mathbf{x}_t} \log p_t(\mathbf{x}_t) \right] dt + \sqrt{\beta(t)}d\mathbf{w}, \tag{15}$$

where $\beta(t)$ is the diffusion rate and $\mathbf{w}$ is the standard Wiener process and $\mathbf{x}_t$ is a representation of the data undergoing the diffusion process. $p_0(\mathbf{x})$ is the distribution of the initial clean data and $p_t(\mathbf{x})$ is the distribution of noisy data corrupted by the forward diffusion process at time $t$.

While the distribution $p_t(\mathbf{x}_t)$ is intractable, the score

$$s_\theta^*(\mathbf{x}_t, t) = \nabla_{\mathbf{x}_t} \log p_t(\mathbf{x}_t), \tag{16}$$

can be approximated by a neural network (e.g. an U-net). This has been shown to be a powerful method for fast, accurate sampling of images from the data distribution. We aim to leverage the denoising diffusion process to generate images sampled from the (natural) image distribution while adhering closely to specific texture statistics.

To this end, we apply gradient guidance per diffusion step as outlined by Chung et al. Chung et al. (2022). Formally, we may define the texture statistics of image $\mathbf{x}$ as $\sigma(\mathbf{x})$, as the output in the analysis pipeline. For synthesis, we wish to sample images with statistics that are near to $\Sigma$. We can model the allowable deviation by some Gaussian noise $\varepsilon$.

$$\Sigma = \eta(\mathbf{x}) + \varepsilon \tag{17}$$

where $\varepsilon \sim N(0, \eta^2)$. Furthermore, we want the images to be drawn from the image distribution $p_0(\mathbf{x})$. This is equivalent to sampling from the joint distribution

$$p_t(\mathbf{x}, \Sigma) = p_t(\Sigma | \mathbf{x}) \cdot p_t(\mathbf{x}). \tag{18}$$

The denoising process is now

$$d\mathbf{x}_t = \left[ -\frac{\beta(t)}{2} \mathbf{x}_t - \beta(t) \cdot (\nabla_{\mathbf{x}_t} \log p_t(\mathbf{x}_t) + \nabla_{\mathbf{x}_t} \log p_t(\mathbf{x}_t, \Sigma)) \right] dt + \sqrt{\beta(t)} d\mathbf{w}. \tag{19}$$

Note that for clean images $\mathbf{x}_0$, we have $\nabla_{\mathbf{x}_0} \log p_0(\mathbf{x}_0, \Sigma) = \frac{\sigma^2}{2} ||\eta(\mathbf{x}_0) - \Sigma||^2$, the same is not true for noisy images $\mathbf{x}_t$:

$$\nabla_{\mathbf{x}_t} \log p_t(\mathbf{x}_t, \Sigma) \neq \frac{\sigma^2}{2} ||\eta(\mathbf{x}_t) - \Sigma||^2. \tag{20}$$

To approximate the above term, we make use of the Tweedie's denoised estimate

$$\hat{\mu}_0(\mathbf{x}_t) = \frac{1}{\sqrt{\bar{\alpha}(t)}} (\mathbf{x}_t + (1 - \bar{\alpha}(t)) \cdot s_{\theta^*}(\mathbf{x}_t, t) \tag{21}$$

where

$$\bar{\alpha}(t) = \prod_{t_i \leq t} \alpha(t_i) = \prod_{t_i \leq t} (1 - \beta(t_i)). \tag{22}$$

k

This allows us to make the approximation $p(\Sigma | \mathbf{x}_t) \approx p(\Sigma | \hat{\mu}_0(\mathbf{x}_t))$. This finally gives us the denoising step as

$$d\mathbf{x}_t = \left[ -\frac{\beta(t)}{2} \mathbf{x}_t - \beta(t) \cdot \left( s_{\theta^*}(\mathbf{x}_t, t) + \frac{\sigma^2}{2} ||\Sigma - \eta(\hat{\mu}_0(\mathbf{x}_t))||^2 \right) \right] dt + \sqrt{\beta(t)} d\mathbf{w}. \tag{23}$$

Effectively, after each unconditioned diffusion step, a step in gradient descent is taken to minimize the statistics loss $\frac{\sigma^2}{2} ||\Sigma - \eta(\hat{\mu}_0(\mathbf{x}_t))||^2$. In our experiments, we have found latent diffusion models such as Stable Diffusion 1.5 to have better performance and efficiency than pixel-space diffusion. In this case, the main adjustment is to calculate the estimated denoised image $\hat{u}_0 \equiv D(\hat{\mu}_0(\mathbf{x}_t))$ from the estimated denoised latent representation $\hat{\mu}_0(\mathbf{x}_t)$ where $D$ is the decoding function (often from an autoencoder) going from latent space to pixel space. The full procedure is outlined as follows

1. From a reference image $u_{\text{ref}}$, calculate target statistics $\Sigma = \eta(u_{\text{ref}})$.

2. Initialize with random noise latent, perform unconditioned diffusion denoising.

3. After each iteration $i$ at timestep $t$ with latent state $\mathbf{x}_t$, calculate the estimated denoised latents $\hat{\mu}_0(\mathbf{x}_t)$.

4. Decode the estimated denoised latent state to the denoised image estimate

$$\hat{u}_0 = D(\hat{\mu}_0(\mathbf{x}_t)).$$

5. Calculate the loss between the statistics of the denoised estimate and the reference image,

$$J = \frac{\sigma^2}{2} ||\eta(\hat{u}_0) - \Sigma||^2$$

6. Auto-differentiate the loss to obtain the gradient $\nabla_{\mathbf{x}_t} J$ and update the latent state a second time:

$$\mathbf{x}_t \leftarrow \mathbf{x}_t - \zeta_t \cdot \nabla_{\mathbf{x}_t} ||\eta(D(\hat{\mu}_0(\mathbf{x}_t))) - \Sigma||^2,$$

where $\zeta_i$ are tunable update steps.

7. Repeat 3-6 until denoising process is completed at $t = 0$.

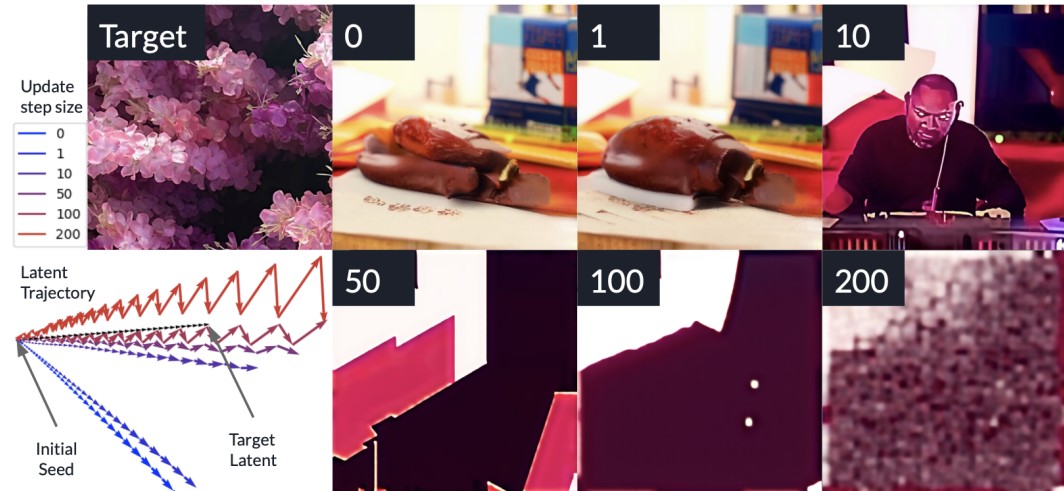

Figure 15: Gradient guided latent diffusion with SD3 over a range of update step sizes. The target reference image is shown on the top-left. The final synthesized images for each step size is shown to the right. In the bottom left, the trajectory of the gradient guided diffusion process is projected into a 2D space (via PCA).

One challenge in implementing gradient guided diffusion is in selecting the update rate.

Shown in Figure 15, we performed gradient guided (latent) diffusion over a range of step-sizes, starting from $\zeta = 0$, unconditioned diffusion. We used the same initial seed and generator to ensure consistency and reproducibility across each run. Initially, without any gradient guidance, an arbitrary, un-related image is generated. The trajectory of the diffusion process, in latent space, is visualized in the bottom left. As the the step-sizes are increased, gradient guidance moves the trajectory closer towards the target – more specifically, the latent representation of the target). However, as the step-size gets larger and larger, the synthesized images start to appear more and more out of distribution.

We hypothesize that this is due to the overall alignment of the denoising step as performed by the diffusion diffusion denoising process and the direction enforced by gradient guidance in matching the statistics. With increasing step size, the latent state is brought further in the direction of denoising than typical at the given timestep. Eventually the accumulate error cause the latents to overshoot the image manifold. Yet, at lower step-sizes, the image generated does not sufficiently match the target statistics.

To overcome this challenge, we implement back-off gradient guidance (BOGG) where the inverse diffusion process is reset partially multiple times before it is run to completion. At each reset, we move from timestep $t_2$ "back" to $t_1$, resetting the scheduler to $\bar{\alpha}(t_1), \alpha(t_1), \beta(t_1)$, etc. We also inject noise to the latent state

$$\mathbf{x}_{t_1} = \sqrt{\frac{\bar{\alpha}(t_1)}{\bar{\alpha}(t_2)}}\mathbf{x}_{t_2} + \sqrt{1 - \frac{\bar{\alpha}(t_1)}{\bar{\alpha}(t_2)}}\xi; \ \xi \sim N(0, 1), \tag{24}$$

running the diffusion process forward from $t_2$ to $t_1$ before restarting the denoising, inverse process again. This way, we're able to have more smaller gradient guidance steps while preventing the denoising process from progressing. See Fig. 16 for an example. The noise schedule of $\bar{\alpha}(t)$ is plotted as a function of the total iterations showing each restart. The gradient guidance continues to minimize the loss during the restarts while the denoising process is held back. The trajectory in latent space is shown on the bottom left where one can see that the iterative denoise, gradient guidance and restart cycle moves the image latent state closer to the target.

On the right of Fig. 16, the finally synthesized picture while generally matching the target statistics, it does not generate a pure texture but rather a framed picture of flowers hang on a flowery wall. This behavior is due to the strong object bias in the training set of SD1.5 as well as most diffusion models.

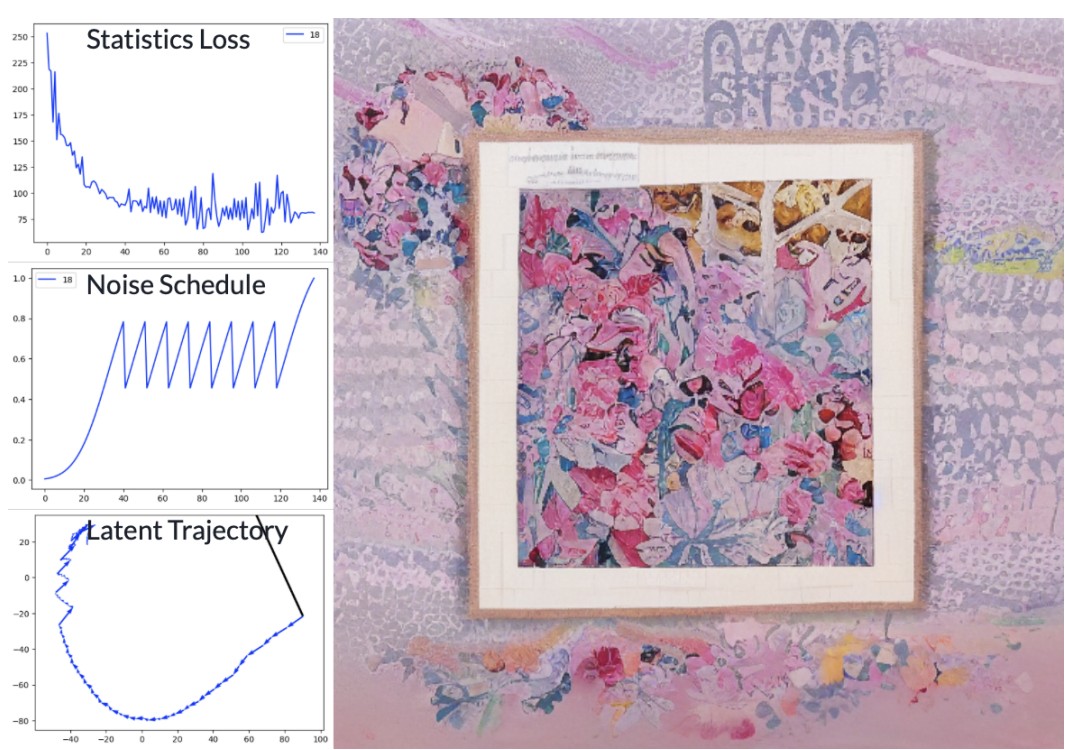

Figure 16: The back-off gradient guidance diffusion. The final generated image is to the right. On the left, we show the gradient guided statistics loss over iterations. The noise schedule of $\bar{\alpha}(t)$ is shown illustrating the restart process. Finally the trajectory in the image latent space is shown at the bottoml

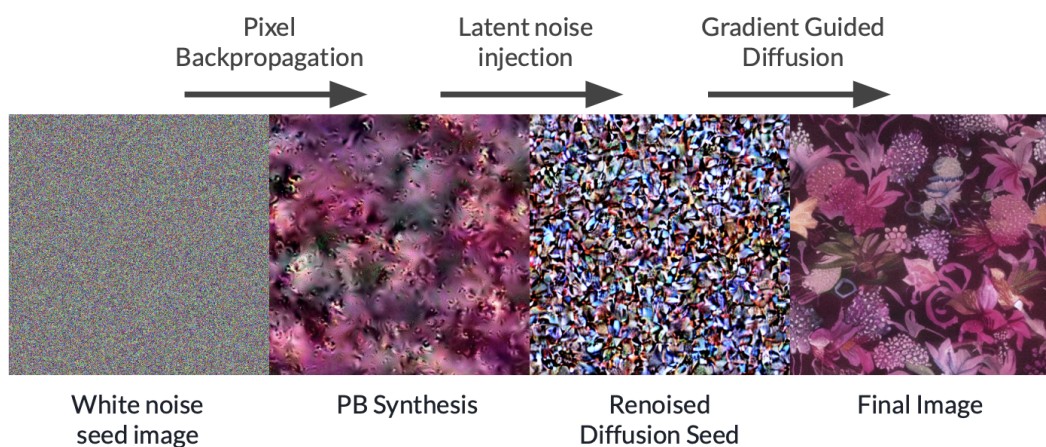

Figure 17: Pixel backprop-gradient guided diffusion pipeline.

We developed another method combining both pixel-backprop and gradient guided diffusion. We start with synthesize through pixel-backprop which lacks any object bias but generates un-naturalistic images. Treating this as a clean image, we then inject noise (see Eq. 24) to bring it to some intermediate timestep. Finally, we run the gradient guided denoising model to completion. An example of this pipeline is show in Fig. 17

