# OpenReview forum: "GramStatTexNet: Efficient, Interpretable, and Neuro-Inspired Spatiotemporal Texture"
_ICLR.cc/2026/Conference — ICLR 2026 Conference Withdrawn Submission_

### Official Review · Reviewer_G5zU · 2025-10-26

**Soundness:** 2
**Presentation:** 1
**Contribution:** 2
**Rating:** 2
**Confidence:** 4

**Summary:**

This paper proposed a texture synthesis method called GramStatTexNet, incorporating the Gram statistics and the handcrafted filter banks. Experiments on image texture, dynamic texture and full image are conducted. Extra guidance of gradient is also considered.

**Strengths:**

As far as I can see, the novelties are:
1. The model combines gram statistics with handcrafted filter banks, then, contrastive training is used to choose the most important features in the Gram matrix.
2. The use of gradient guidance in diffusion model for texture synthesis.

**Weaknesses:**

1. The contributions are incremental.
    1. The author claims that one advantage of this work is that the use of filter banks is neuroscience-informed. (Line 142-144). However, in turns of biology (like Portilla's work), the investigation on the connection with human vision is lacking.
    2. On the other hand, in turns of machine learning, the qualitative performance of the method is rather bad (Fig. 2), and no quantitative comparison is made.
    3. The author also claims the interoperability and efficiency of the method as advatanges compared to the network-based methods like Gatys' work. However, under this standard, the comparison with the classic texture model, like the Gaussian model [1], is necessary but is completely missingg. In additiion, the author also uses networks-based method like SD3, which seems to contradict the claim of interoperability and efficiency.

2. The paper is hard to read due to the lack of clearness. Most of details are hidden in the unorganized material in the appendix. Just to name a few issues.
    1. Line 265. Fig reference empty.
     2. Line 1211. What is this k?
     3. Line 1184. What is $\Sigma$? is it $\sigma$?
     4. Line 1188. What is $\eta$? It is a function of $x$ in Eqn. 17, but then it becomes the standard deviation in this line.
     5. Line 1212. What is this approximation? What does it mean?

3. The lack of comparison. In Fig.4, no Portilla's, or Gatys' results are shown. For dynamic texture and peripheral no baseline is shown at all. In the gradient guided method, no quantitative result is shown.

4. The gradient guided results (Fig. 8) do not look like the input. How to evaluate the result? Is it still a texture synthesis result? There is no explaination of the result.

[1]  Gui-Song Xia, Sira Ferradans, Gabriel Peyré, and Jean-François Aujol Synthesizing and Mixing Stationary Gaussian Texture Models

**Questions:**

1. The title is kind of wired. Is there a "model" or "synthesis" missing at the end of the title?
See weakness for other questions.

---

### Official Review · Reviewer_NyVW · 2025-10-28

**Soundness:** 1
**Presentation:** 1
**Contribution:** 2
**Rating:** 2
**Confidence:** 5

**Summary:**

The paper introduces GramStatTexNet, an analysis-by-synthesis model designed to bridge the gap between interpretability, biological plausibility, and high performance in texture and peripheral vision modelling. The challenge lies in creating sophisticated texture models. The authors state that classical, biologically-inspired models are interpretable but slow and limited to spatial textures, while modern deep learning models are fast and generate high-quality results but are over-parameterized and lack interpretability or biological grounding. GramStatTexNet attempts to address this by combining the multi-scale Gabor filter structure of classic models with the power of Gramian-based approaches (filter correlation statistics). Key contributions include achieving synthesis quality similar to deep learning models while remaining efficient and interpretable, using contrastive learning to identify a significantly reduced set of statistics that retains high-quality synthesis, extending the model to full-frame peripheral vision synthesis through spatial pooling, and extending the framework to the spatiotemporal domain for video texture synthesis.

The authors conducted several experiments to validate GramStateTexNet. For spatial texture synthesis, their model showed qualitative improvement over previous multi-scale-pyramid methods (like Portilla & Simoncelli) and comparable quality to larger, neural-network-based methods (like Gatys et al.), despite using an order of magnitude fewer parameters. A contrastive learning experiment was used to reduce the full set of 28,929 statistics, identifying a top-5,000 subset that still qualitatively outperformed the Portilla & Simoncelli model, achieving an 83% parameter reduction. For peripheral vision modelling, they incorporated spatial pooling to create foveated synthesis and showed that the reduced statistics set could achieve a greater than 50% compression compared to the input image while maintaining good results. Furthermore, the paper demonstrates the model's application to spatiotemporal texture by designing a human-vision-informed spatiotemporal filter bank and generating synthetic texture videos that visually match the target's spatiotemporal texture statistics. Finally, they explored guiding modern diffusion models using their statistics, developing a pixel backprop-gradient guidance method to produce photorealistic, style-matched textures that minimize the diffusion model's object bias.

**Strengths:**

- Originality
    - There is originality in using multi-scale oriented Gabor filters as the sole means of capturing texture statistics via the Gramian.
- Quality
    - N/A
- Clarity
    - The overarching approach to the texture modelling is fairly straightforward and easy to understand from a high level, not including the parameter reduction component or diffusion experiments.
- Significance
    - N/A

**Weaknesses:**

- Originality
    - The authors are missing references to previous work [1, 2, 3] which have managed to successfully describe dynamic textures using a perceptually-grounded, human-vision-based approach (i.e., multi-scale oriented spacetime Gabor filters). This goes against the authors' statement of being the first to create such a model.
    - The authors are missing a citation to Tesfaldet et al's work on dynamic texture synthesis [3], being the first deep-learning based model designed for the task.
- Quality
    - Missing comparisons with muNCA [4] and DyNCA [5] in terms of compressed texture representations, with the former being for spatial textures and the latter being for spacetime textures.
    - The quantitative evaluations and comparisons on spatial texture synthesis (and spacetime texture synthesis) are lacking. For spatial texture synthesis, only a few models are compared against, missing models such as muNCA [4] (in terms of comparing parameter count vs. quality). For spacetime texture synthesis, no models are compared against. One can compare against Two-Stream [3] and DyNCA [5].
- Clarity
    - A lot of references to tables and figures in the appendix (e.g., L203-215). If these are important, they should be placed within the main manuscript.
    - Contrastive learning objective isn't well-explained, lacking equations (i.e., the loss function, inputs/outputs) to show exactly what's going on.
    - Some figure captions are relatively unclear and/or contain errors (e.g., Fig 2, Fig 4).
    - The implementation of the diffusion experiments are not clearly explained, making it difficult to appreciate results.
    - Appendix is missing sections, which contributes to the overall rushed feeling of the paper.
- Significance
    - It's difficult to gauge the impact of the peripheral synthesis experiments, as the results are fairly limited.
    - A human-vision-grounded dynamic texture model already exists [1, 2, 3], which eliminates one of the paper's contributions, reducing its significance.
    - It's difficult to gauge the impact of the spatial and spacetime texture synthesis results due to a lack of extensive quantitative evaluations and comparisons with other approaches. A user study (forced choice evaluation) might be in order here.
    - The spacetime texture synthesis results do not look impressive compared to previous approaches such as [3, 5].

[1] Konstantinos G. Derpanis and Richard P. Wildes. Dynamic Texture Recognition based on Distributions of Spacetime Oriented Structure. In Proceedings of the IEEE/CVF Conference Computer Vision and Pattern Recognition (CVPR), 2010.

[2] Konstantinos G. Derpanis and Richard P. Wildes. Spacetime Texture Representation and Recognition Based on a Spatiotemporal Orientation Analysis. IEEE Transactions on Pattern Analysis and Machine Intelligence (PAMI) 34 (6), 1193-1205, 2012.

[3] Mattie Tesfaldet, Marcus A. Brubaker, and Konstantinos G. Derpanis. Two-stream Convolutional Networks for Dynamic Texture Synthesis. In Proceedings of the IEEE/CVF Conference on Computer Vision and Pattern Recognition (CVPR), 2018.

[4] Alexander Mordvintsev and Eyvind Niklasson. muNCA: Texture Generation with Ultra-Compact Neural Cellular Automata. arXiv preprint, 2021.

[5] Ehsan Pajouheshgar, Yitao Xu, Tong Zhang, and Sabine Süsstrunk. DyNCA: Real-Time Dynamic Texture Synthesis Using Neural Cellular Automata. In Proceedings of the IEEE/CVF Conference on Computer Vision and Pattern Recognition (CVPR), 2023.

**Questions:**

1. Due to one of the contributions being the compressed nature of the texture representation, how does your approach qualitatively compare with other compressed texture representations like muNCA and DyNCA?
2. Can you comment on the relationship between your spacetime texture representation and that introduced by Derpanis et al. [1, 2] and Tesfaldet et al. [3]? Specifically, under the context of your statement of being the first to create a perceptually-grounded, human-vision-based texture model for video.
3. Can you explain the inputs/outputs and loss for the contrastive learning objective?

---

### Official Review · Reviewer_kJZD · 2025-10-29

**Soundness:** 3
**Presentation:** 3
**Contribution:** 2
**Rating:** 4
**Confidence:** 4

**Summary:**

The paper proposes GramStatTexNet, a biologically inspired texture synthesis model that merges multi-scale Gabor pyramid filters with Gram-matrix correlations. It achieves deep-learning–level synthesis quality while remaining interpretable and efficient. Using contrastive learning, the authors identify key statistics for texture discrimination, enabling major parameter reduction. The model extends to peripheral vision (foveated image synthesis) and spatiotemporal textures (video modeling) and explores integration with diffusion models. Overall, it offers an interpretable and efficient bridge between neuroscience-based and deep-learning texture synthesis.

**Strengths:**

- The paper tackles an important problem in computer vision and graphics, with a novel approach.
- The model is interpretable and biologically grounded, combining vision-based pyramid filters with modern gram-based representations.
- The model achieves comparable quality with some deep learning models, with many less trainable parameters.
- The contrastive analysis in this paper provides a principled way to identify which statistics matter most for texture discrimination.
- The spatiotemporal textension introduces a new biologically plausible model for dynamic texture synthesis-
- Evaluation is somewhat comprehensive.

**Weaknesses:**

- The video results are not evaluated quantitatively in any meaningful way, analysis is limited to some qualitative analysis.
- The diffusion guidance part seems very preliminary.
- The method seems focused on very stochastic types of textures, thereby missing more complicated, human-made textures.
- There is no human perceptual validation.
- Contrastive learning lacks ablation of hyperparameters or generalization tests.
- The contribution seems incremental in the scientific sense, and it is more of an engineering effort of combining existing ideas into a new pipeline.
- The paper is dense, long, and very heavy on implementation details, taking away from readability.
- The paper does a very limited job in its contextualization in the literature of texture synthesis, missing many important works in the generative models-for-texture space. Please see TexTile: A Differentiable Metric for Texture Tileability (CVPR24) for a review on the topic.

**Questions:**

- How does the model behave in more structured textures (eg fabrics)?

---

### Official Review · Reviewer_XoUB · 2025-10-30

**Soundness:** 2
**Presentation:** 2
**Contribution:** 3
**Rating:** 2
**Confidence:** 3

**Summary:**

The paper proposes GramStatTexNet, a texture model that replaces hand-curated statistics in classical pyramid models with a Gram matrix over responses of a biologically inspired multi-scale Gabor/steerable-pyramid filter bank. Concretely, an image is decomposed into “pyramid images” (by scale, orientation, color, phase), and the upper triangle (incl. diagonal) of the inter-channel correlation matrix forms the statistics vector. Synthesis optimizes pixels (initialized randomly) to match a target’s statistics. The authors introduce a contrastive learning module (a single FC layer trained with InfoNCE) that ranks statistic importance for categorizing textures and show that optimizing on the top-N ranked statistics retains synthesis quality. They demonstrate foveated peripheral syntheses and extend the approach to videos by designing a spatiotemporal Gabor bank (tiling spatial and temporal frequency) and matching Gram statistics over time. Finally, they explore gradient-guided diffusion for texture synthesis using pre-trained models (SD1.5) proposing back-off gradient guidance (BOGG) and pixel-backprop + gradient guidance (PB-GG) to improve texture statistics adherence.

**Strengths:**

The paper introduces an original novel texture synthesis method that offers an interpretable alternative to deep features with comparable visual quality. Furthermore the authors introduce a contrastive ranking of statistics that transfers to synthesis quality, enabling an 83% parameter reduction while retaining quality.

**Weaknesses:**

While the paper core idea has indeed some merit, the overall impression is that the current form of the manuscript still requires substantial work before reaching acceptance level: for example, some sections of the Appendix are incomplete (e.g. A.3, A.4, A.5, A.8 etc...) with just the heading but with no content, or some images are hard to interpret/poorly formatted (e.g. Figure 11 y tick labels).

Apart from these formatting/writing issues, there are several important core issues with the work.

1. The paper introduces a novel technique (GramStatTexNet) but offers no quantitative comparisons with existing methods in the literature (e.g Gathys et al. 2015, Ulyanov et al. 2017). This is especially relevant as in Figure 2 qualitative comparisons are reported where a drop in visual quality between the authors' technique and Gathys et al. 2015 can be seen. While this can only be partially justified by the significant smaller number of parameters, a quantitative comparison (e.g. LPIPS, SSIM etc...) is needed to asses the technique's worth.
2. The paper motivates peripheral models by human perception but does not include human studies (cited as future work). Since metamers are defined psychophysically, a claim that this new technique can effectively generate such visual examples (L325-327) appears unsubstantiated.
3. The video results are again poorly supported by the analysis: superior performance against a frame-by-frame method (perhaps with some trivial linear interpolation) is claimed (L411) but not shown, comparison against style-transfer video model (which the author cite in the Related Work section) are not presented.
4. The section about diffusion model lacks important quantification on the proposed new methods: e.g. PB-GG performs PB after a given amount of steps and then starts the diffusion process at an intermediate point, what is the influence of these two choices (how many PB steps/where to start the denoising process) on the final result? What is the tradeoff between statistic-loss and FID for example?
5. The author report quantitive metrics (Figure 4) for their texture synthesis but fail to discuss some curious features, such as the non-monotonic behaviour of the top-N feature drop (Is using 5k feature better than using 10k? Are those differences even meaningful? Can the author provide confidence interval for those values?), or the fact that random depletion seems to be preferable to top-N according to the LPIPS metric.

----------------
### Typos:
- L256: repeated "the"
- L265: missing reference, i.e. Figure (??)
- L293: double dot end of sentence
- L1211: k

**Questions:**

On top of the core issue mentioned in the Weakness section, we propose the following questions to the authors:

1. Why do you chose SGD optimisation for the contrastive FC model while an analytical solution was available? Could you provide a numerical experiment comparing the analytical solution’s embedding to SGD-trained FC across several training sizes?
2. How sensitive are video results to the choice of filter placements and size?
3. How robust is the importance ranking to family size? If you subsample large families to equalize counts and retrain the FC layer, do the same families remain dominant?

---

### Note · Authors · 2025-12-23

**Comment:**

We withdraw this submission.

**Withdrawal Confirmation:**

I have read and agree with the venue's withdrawal policy on behalf of myself and my co-authors.